# Polarity-JaM: an image analysis toolbox for cell polarity, junction and morphology quantification

Wolfgang Giese ●[1,2,8] ✉, Jan Philipp Albrecht ●[1,3,4,8], Olya Oppenheim ●[1,2,5], Emir Bora Akmeriç ●[1,2,5], Julia Kraxner[1,2], Deborah Schmidt[1,3], Kyle Harrington ●[1,6] & Holger Gerhardt ●[1,2,5,7]

Cell polarity involves the asymmetric distribution of cellular components such as signalling molecules and organelles within a cell, alterations in cell morphology and cell-cell contacts. Advances in fluorescence microscopy and deep learning algorithms open up a wealth of unprecedented opportunities to characterise various aspects of cell polarity, but also create new challenges for comprehensible and interpretable image data analysis workflows to fully exploit these new opportunities. Here we present Polarity-JaM, an open source package for reproducible exploratory image analysis that provides versatile methods for single cell segmentation, feature extraction and statistical analysis. We demonstrate our analysis using fluorescence image data of endothelial cells and their collective behaviour, which has been shown to be essential for vascular development and disease. The general architecture of the software allows its application to other cell types and imaging modalities, as well as seamless integration into common image analysis workflows, see https://polarityjam.readthedocs.io. We also provide a web application for circular statistics and data visualisation, available at www.polarityjam.com, and a Napari plug-in, each with a graphical user interface to facilitate exploratory analysis. We propose a holistic image analysis workflow that is accessible to the end user in bench science, enabling comprehensive analysis of image data.

Cellular polarity is important in many biological phenomena, spanning from developmental processes such as angiogenesis to tissue repair in the adult organism. Cell migration, cell division, and morphology depend on prior polarisation and breaking of spatial symmetry. Spatial reorganisation of the plasma membrane, cytoskeleton, cell-cell junctions, or organelles is required to establish an axis of polarity with a distinct 'front and back' direction, in order to guide directed processes[1]. In these processes, cells react and adapt according to multiple and often conflicting cues from their environment[2].

Fluorescence microscopy has become an invaluable tool for producing high-resolution, high-content images of in vitro systems as well as in vivo tissues. These images can be obtained at a subcellular resolution of less than one micron, with multiple fluorescence channels acquired in parallel, often on multiple planes, allowing for detailed quantification of cellular polarity and asymmetries. At the same time,

[1]Max Delbrück Center for Molecular Medicine in the Helmholtz Association, Berlin, Germany. [2]DZHK (German Center for Cardiovascular Research), Berlin, Germany. [3]HELMHOLTZ IMAGING, Max Delbrück Center for Molecular Medicine in the Helmholtz Association, Berlin, Germany. [4]Faculty of Mathematics and Natural Sciences, Humboldt-Universität zu Berlin, Berlin, Germany. [5]Charité - Universitätsmedizin Berlin, Berlin, Germany. [6]Chan Zuckerberg Institute for Advanced Biological Imaging, Redwood City, CA, USA. [7]Berlin Institute of Health, Berlin, Germany. [8]These authors contributed equally: Wolfgang Giese, Jan Philipp Albrecht. ✉e-mail: wolfgang.giese@mdc-berlin.de

deep learning segmentation algorithms have developed at a staggering pace over the past few years, enabling the segmentation of individual cells and organelles with near-human accuracy[3–9]. This opens up a wealth of unprecedented possibilities, but also creates new challenges for comprehensible and interpretable image data analysis workflows that fully exploit these new potentials.

Each cell has a unique shape, a particular spatial distribution of organelles, and contains different absolute amounts and distributions of protein species, which are measured by intensity and gradients, respectively. In addition, junctional cell-cell contacts exist with their own morphological phenotypes[10,11]. Taken together, this image-based information provides a snapshot of a cell's state, which we aim to turn into quantitative and comparable features. We demonstrate our investigations on image data from endothelial cells (ECs). ECs line the inside of blood vessels and play a crucial role in organ function and health of the whole organism. Migration of these cells is important for vessel formation and repair, but can also be involved in disease processes in the cardiovascular system, cancer, or inflammation. ECs are

sensitive to shear stress when blood flow passes through blood vessels, causing alterations at the collective and single-cell level, including morphological changes in ECs alignment, shape, size, as well as subcellular changes to cell-cell junctions, organelle polarity, and gene expression. The location or distribution of organelles in the cell can be quantified in relation to the nucleus or the cell centre. For example, the position of nucleus and Golgi apparatus are used to compute nuclei-Golgi polarity, which has been positively correlated with directed motility[12] and cell orientation[13] in several cell types, including epithelial cells and ECs, see Fig. 1A, B. EC polarity can be induced by multiple signaling cues, including shear stress and VEGFA, which can be in competition with each other. Various EC shear stress sensors have been identified that induce EC polarisation, including Piezo1, plexinD1, focal adhesions (FA), the VEGFR2/PECAM1/VE-cadherin complex, or caveolae[14]. As a result, ECs adapt their shape and orientation to shear forces[13,15]. Cell shape is often measured using circularity, shape index, or length-to-width ratio (LWR). While LWR has been positively correlated with VEGFA treatment in all EC types[16], the collective signature

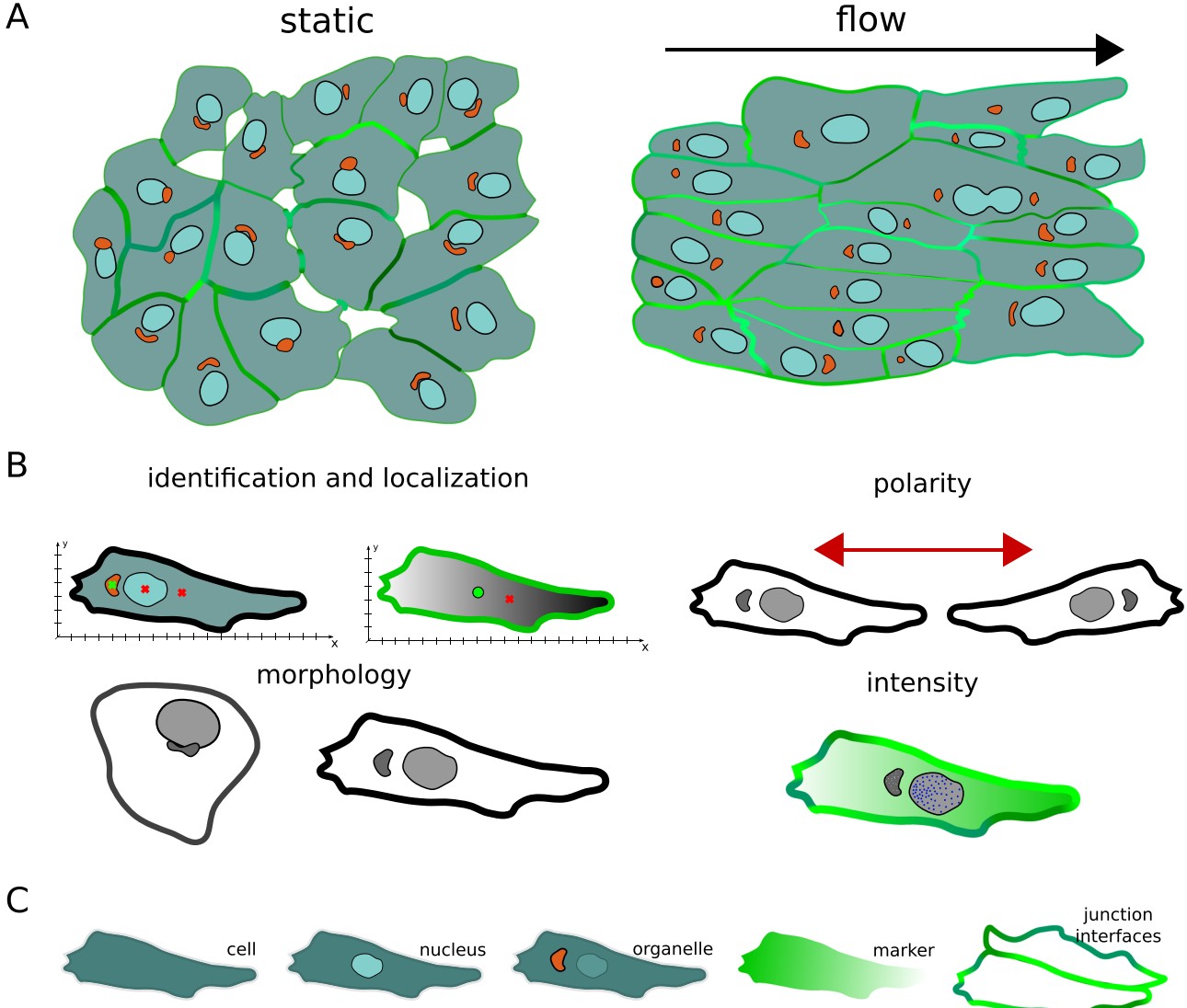

**Fig. 1 | An overview of feature categories and targets. A** Endothelial cells under static conditions exhibit a random polarity, while endothelial cells exposed to laminar flow polarise against the direction of flow (indicated here by the relative nuclei-Golgi polarity) and elongate. The flow direction is indicated by the black arrow. **B** Categories of image-based features comprise identification and localisation, polarity, morphology, and intensity readouts, which can be applied to (**C**)

several targets, including cell, nucleus, organelle, marker and junction interfaces. For example, localisation can be computed for the targets cell, nucleus, and organelle. Similarly, morphology features describing the elongation of an object can be calculated for different targets, such as the cell and nucleus. Polarity-JaM features are generated automatically based on available targets and user configuration. See Supplementary Tables 1 and 2 for an overview.

also differs across organs and microenvironments[17]. ECs exposed to undisturbed flow are elongated with an increased LWR and are aligned with the direction of flow, whereas ECs in areas of disturbed flow are more cuboidal and are randomly oriented, both in vivo and in vitro[18].

A major challenge in the creation of every image analysis pipeline is identifying and measuring informative features. This search has a large iterative component and is based on precise and accurate measurement of the relevant microscopy data[19]. Several studies have proposed meaningful measures such as Quantify Polarity[20], Junction Mapper[11] and Griottes[21] or reviewed existing ones[19]. However, it is not yet possible to integrate these different aspects into a single pipeline and perform a multivariate analysis. For a comparison of tools, we refer to Supplementary Table 3. These requirements and constraints motivated the development of the Polarity-JaM package, which is built to streamline the process of exploratory image analysis; this is accomplished by providing the end user with functionality that includes a wide range of features, explanatory metadata, clear and concise documentation, and high testing coverage. Proper meta-data and testing ensure that parameters can be tuned safely and systematically, and the functionality of the main components of the package can be -reasonably- extended to new types of analysis. All relevant analysis can be performed with our package, making installation and usage straightforward.

To cover the wide range of properties that can be extracted from multichannel images, we have developed a framework that makes them easy to find and explain. We therefore introduce different feature categories, including (1) object identification and localisation, (2) morphology, (3) polarity, and (4) intensity-related properties, see Fig. 1B. Each of these categories can be applied to different targets, including single cells, nuclei, organelles, the intensity profile of a marker, and junction interfaces, see Fig. 1C. For example, localisation can be applied to the cell, nucleus and organelle. In the same way, we can calculate morphology features for different targets, e.g. the LWR can be extracted from a cell and nucleus. Polarity features are generally based on (a) differences between the left and right sides of a cell, (b) the calculation of an orientation angle with respect to an axis, here called axial polarity, or (c) the calculation of a directional 'front-back' polarity, see Fig. 1B and Supplementary Table 1. Note that in the latter case, polarity features can have two targets; in the example of nuclei-Golgi polarity, the positions of the nuclei (target 1) and Golgi (target 2) relative to each other are extracted, which is used to define the 'front-back' polarity of a cell with a defined direction. We will discuss these examples in the Results section. Depending on the input image and the configuration, different features and combinations will be computed.

In this article, we present Polarity-JaM, an open-source software suite for measuring and analysing cellular properties in microscopic images, using EC biology as an example. To visualise and explore the wide range of cellular polarity, morphology and junctional features, we have developed an R-shiny application that integrates bespoke statistical tools for circular data not available in standard toolboxes. We introduce informative statistical plots, including circular histograms, polarity indices and confidence intervals, to display all relevant information. For terminology used in our article, we provide a glossary at the end of the supplement, see Supplementary Table 15. In summary, we propose a holistic image analysis workflow that is accessible to the end user in bench science, enabling comprehensive analysis of collective cell systems.

## Results

### Asymmetries in subcellular localisation of organelles

Cell polarisation is a dynamic process that involves the reorganisation of various cellular components, including the cytoskeleton, intracellular signaling molecules, organelles, and the cell membrane. The nucleus is the most prominent organelle in eukaryotic cells and is constantly exposed to intrinsic and extrinsic mechanical forces that trigger dynamic changes in nuclei morphology and position[22]. Organelles like the centrosome and Golgi apparatus are located outside the nucleus and typically not at the cell's geometric centre[23,24]. Their positions are crucial for various directed processes, such as cell division, migration, adhesion, and cell-cell contact formation. The positioning of these organelles can have passive or active effects on mechanisms such as Rho signaling and Map Kinase signaling[25].

We exemplified our approach using nuclei-Golgi positioning and nucleus displacement as read-out; see Fig. 2. Sprouting angiogenesis and vascular remodelling are based on directional migration of ECs[26]. In response to flow-induced shear stress, the position of the Golgi apparatus is relocated upstream of the nucleus, against the flow direction[13]. Nuclei-Golgi polarity is often used as a proxy for the migration direction in static images in vivo and in vitro. We applied shear stress levels of 6 dyne/cm² and 20 dyne/cm² for 16 h under different media conditions, to induce robust collective polarisation of the EC monolayers (see Fig. 2). Microscopic images contained a junction channel, a nuclei channel, and Golgi staining, see Supplementary Fig. 1A and Supplementary Fig. 2 for an example. Using the Cellpose[5] algorithm, we obtained segmentations for cells and nuclei. Golgi segmentations were obtained by applying Otsu thresholding directly to the Golgi channel and superimposing the resulting mask with the cell segmentation from Cellpose, see Supplementary Fig. 1B for an example. The nuclei-Golgi vectors were automatically calculated for each cell, see Fig. 2A.

The collective strength of polarisation is commonly measured using the polarity index[27], which is calculated as the resultant vector of all orientation vectors from each single cell. Mathematically we obtain a vector for every single cell

$$\mathbf{r}_i = \begin{pmatrix} \cos \alpha_i \\ \sin \alpha_i \end{pmatrix}. \tag{1}$$

Note that $\alpha_i$ in this example is the orientation of the displacement from nucleus to Golgi, but can be a placeholder for any given directed 'front-rear' polarity feature, including nuclei displacement with respect to the cell centroid and others, see Supplementary Table 1. The average of the individual vectors is used to calculate the resultant vector from

$$\mathbf{r} = \frac{1}{N} \sum_{i=1}^{N} \mathbf{r}_i, \tag{2}$$

where $N$ is the number of cells. The length of this vector, computed from $R = \|\mathbf{r}\|$, is called the polarity index and its direction is the mean of the distribution. The value of the polarity index varies between 0 and 1 and indicates how much the distribution is concentrated around the mean direction. A polarity index close to 1 implies that the data are concentrated around the mean direction, while a value close to 0 suggests that the data are randomly distributed or spread in several directions. In summary, the polarity index indicates the collective orientation strength of the cell layer or tissue. Note that the polarity index is closely related to the variance of the distribution by $S = 1 - R$. In Fig. 2C, E, F the value of the polarity index is shown and indicates the strength of the collective flow response.

To complement this analysis, we introduce a signed polarity index (V), which is derived from the V-test statistics[28,29] and assumes a known predefined polar direction as a reference. For example, if we assume that flow is orientated from left to right, we set the polar direction to an angle of $\alpha_p = 0$ in our reference system, see Fig. 2C. The signed polarity index is computed from:

$$V = cR, \tag{3}$$

$$\text{with } c = \cos(\bar{\alpha} - \alpha_p). \tag{4}$$

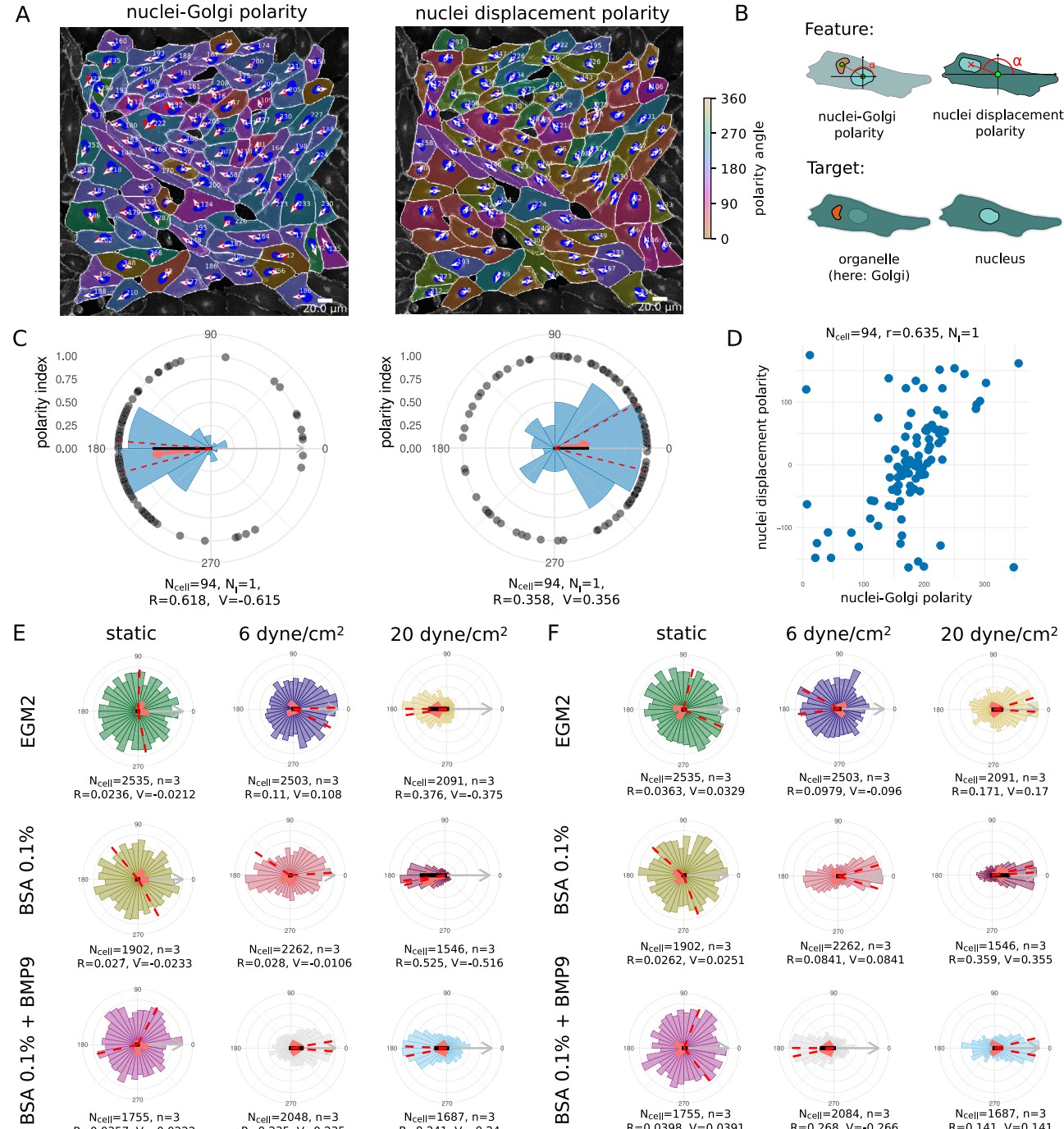

**Fig. 2 | Asymmetric localisation of organelles as a measure for cell polarity.**
**A** Human umbilical vein endothelial cells (HUVECs) in culture exposed to 20 dyne/cm² shear stress for 16 h. Nuclei-Golgi polarity [left] and displacement of the nuclei within each cell with respect to its centre [right]. Orientation is indicated by a cyclic colour scheme and white arrows from the centre of the nuclei to the centre of the Golgi (nuclei-Golgi polarity) or from the centre of each cell to the nucleus (nucleus-displacement polarity). The flow direction is always from left to right. Scale bar 20 μm. **B** Schematic representation of the features and target for this figure. A full overview of features that can be extracted from nuclei and organelle targets is given in Supplementary Tables 4 and 5. **C** Circular histograms of the distribution of cell orientations of a single image; the red arrow indicates the mean direction of the cell collective and its length is the polarity index. The red dashed lines indicate the 95% confidence intervals. Black dots indicate single-cell measurements. The grey arrow indicates the polar vector that points in the direction of flow, and the length of the black bar indicates the signed polarity index (V). **D** Circular correlation of nuclei-Golgi polarity and nuclei displacement polarity. **E** Ensemble plot of nuclei-Golgi polarity generated with the Polarity-JaM app for different flow conditions.
**F** Ensemble plot of the nuclei displacement orientation polarity generated with the Polarity-JaM app for different flow conditions. Source data are provided as a Source Data file.

The signed polarity index varies between −1 and 1 and indicates the strength of polarisation with respect to the polar direction. In our example, a value of −1 indicates that all cells are perfectly oriented against flow, while a value of 1 indicates that all cells are perfectly orientated with flow. For values in between, the distribution is more spread or diverges from the polar direction. Therefore, the signed polarity index measures both the deviation from given polarity direction and the spread of the distribution. Note that the polar direction

provides a reference for comparing conditions, therefore we also calculate the V-score for the static condition, even though there is no flow. We suggest a graphical design for the representation of the polarity index, the signed polarity index, the polar direction, confidence intervals, circular histograms, and single measurements, see Supplementary Fig. 3.

To compare and correlate nuclei-Golgi polarity with nuclei displacement, we repeated the same calculations, but with polarity vectors from cell centres to nucleus centres, see sketch in Fig. 2B. The distribution of the polarity index (R) and signed polarity index (V) for $N_i = 30$ images per condition is shown in Supplementary Fig. 4A, B for the nuclei-Golgi polarity and in Supplementary Fig. 4C, D for nuclei displacement. We found that the nuclei-Golgi polarisation of ECs under shear stress is highly correlated with the displacement of the nuclei and points in the opposite direction, Fig. 2C, D. We found the same behaviour for a wide range of flow conditions; see Fig. 2E, F and Supplementary Fig. 5, but with different magnitudes depending on the media conditions and magnitude of shear stress. Our web application includes a number of different statistical tests, including the Rayleigh test, the V-test, the Watson test, and Rao's spacing test, see Supplementary Note 5. It should be noted that these tests are not appropriate for measurements of individual cells, given that the groups of single cells within a monolayer or tissue exhibit significant correlation. We therefore recommend the use of estimation statistics[30] to calculate effect sizes of collective parameters such as the polarity index (R) and signed polarity index (V) (See Supplementary Fig. 4).

In[31] a systematic analysis has demonstrated that the V-test, which is based on the computation of the signed polarity index, is recommended over other tests if an expected direction is known a priori. Note that here we use the flow direction as the polar direction, whereas for the V-test the expected polarisation direction must be used. For the nuclei-Golgi polarity, for example, the expected polarisation direction points in the opposite direction to the flow direction, while for the nuclei displacement both the flow direction and the expected polarisation direction are the same. The signed polarity index in our data provides a more accurate distinction of all conditions with respect to control, in particular for 6 dyne/cm² and 20 dyne/cm². In conclusion, it is advantageous to use the signed polarity index when the expected direction of polarisation or the direction of an external signal is known in advance.

## Shape orientation and morphology

Cell morphology is an essential part of cell biology as it provides insights into the structure and function of cells and can be used to understand the effects of different treatments, genetic mutations or identify therapeutic targets. The shape of the cell also affects the ability of cells to interact with their environment and to respond to external signals. Cell protrusions can function as small pockets and reaction chambers, while signaling gradients are more easily stabilised along the long axis of the cell than the short axis[32,33]. Cell shape is also coupled with the migration direction; for example, in keratocyte-like motion, polarisation occurs along the short axis of the cell[34], while other cell types migrate along the long axis. Furthermore, cell shape can be an indicator of the mesoscale properties of tissue or cell monolayer and is often used as an order parameter in soft matter physics[35,36]. In summary, cell shape and orientation are important readouts for directed cellular processes.

Recent studies have shown that EC monolayer orientation is modulated by both physical and chemical stimuli[37,38]. In particular, laminar shear stress induces collective EC alignment and elongation. We compared the morphological response of ECs under three experimental conditions: static, 6 dyne/cm² shear stress at 24 h and 20 dyne/cm² shear stress at 48 h, see Fig. 3. We determined cell orientation from the angle of the major axis to the x-axis (Fig. 3C, D), which results in axial orientation data for each single cell. Note that

angular data of cell shape orientation are referred to as axial data, which means that all orientation angles $\phi_i, 1, \cdots, N$ take values between 0 and 180°. Thus, we do not distinguish the front and back of the cell (or nucleus), see Fig. 3G. Circular histograms showing the angular distribution were generated with the Polarity-JaM app. It is important to note that all axial orientation measurements have a periodicity of 180° and are therefore repeated every 180° in the circular histograms. The duplicated data points are therefore shown transparent, see Fig. 3E and Supplementary Fig. 3. To compute statistical quantities, these axial orientation data were converted to directional data by doubling all values $\theta_i = 2\phi_i$. The mean direction was calculated from $\bar{\phi} = \frac{\bar{\theta}}{2}$, where $\bar{\theta}$ is the common circular mean of the directional data $\theta_i$. Similarly, the polarity index was calculated as the length of the mean resulting vector of directional values $\theta_i$. Again, the polarity index varies between 0 and 1 and indicates how much the distribution is concentrated around the mean. A polarity index close to 1 implies that the data are concentrated around the mean direction, while a value close to 0 suggests that the data are evenly distributed or random. The V-score can be computed in the same fashion as for directed circular data (see Supplementary Note 5 for more details).

Applying those measures, we found a strong response in the orientation parallel to the flow at 6 dyne/cm² and 20 dyne/cm² with a V-score of 0.507 and 0.51 respectively. For static condition, there is no collective orientation with respect to flow indicated by a V-score of −0.0408, which is close to zero. We repeated the same computation for nuclei orientation and found strong correlation with cell orientation, but with overall less concentrated collective orientation of the nuclei, Supplementary Fig. 6E, F. Elongation was characterised by the ratio of length to width, which where computed from the repsective image moments. We observed only a weak response in elongation for 6 dyne/cm², but a strong response for 20 dyne/cm² with a length to width ratio greater than 3 (Fig. 3F). To complement this analysis, we further quantified the shape symmetry with respect to the cue direction by splitting each cell along an axis perpendicular to the flow direction through its centre, mirroring each half, and calculating its shape symmetry based on the intersection over union of the resulting areas, see Supplementary Fig. 6C. This symmetry score ranges from 0, meaning very asymmetric, to 1, meaning perfectly symmetric. We found that the EC monolayer at high shear stress of 20 dyne/cm² appears to be more swirly than at 6 dyne/cm², resulting in a much lower symmetry score. In addition, the cell shape appears more curved at high shear stress. Note that shape symmetry is not captured by either elongation or orientation. For a complete list of features with target cell, please refer to Supplementary Table 6.

## Quantification of intracellular signalling gradients

Gradients or asymmetric distributions of signaling molecules are inherent in cell polarity. Often these asymmetries are decoded into rather small and subtle gradients that can be amplified by signaling feedback systems, such as the Rho GTPase system[39]. The establishment of these signaling gradients within single cells allows cell collectives to respond to their environment in a coordinated manner and is used to control cell migration, differentiation, and other cellular processes[40]. Quantifying the direction and strength of intracellular signaling gradients between different experimental conditions is therefore crucial to gain insight into the underlying processes.

The notch signaling pathway is involved in the regulation of various genes responsible for angiogenesis[41] and has been shown to be a mechanosensor in adult arteries[42]. Therefore, the effects of this pathway are of great interest. We present an automatic quantification approach of signaling gradients for each cell with the example of the NOTCH1 protein using our tool. We investigated both circular and linear features. First, the marker polarity is a circular feature, which can be described as the direction from the geometric cell centre to the weighted centre of marker intensity of the cell. Second, we computed

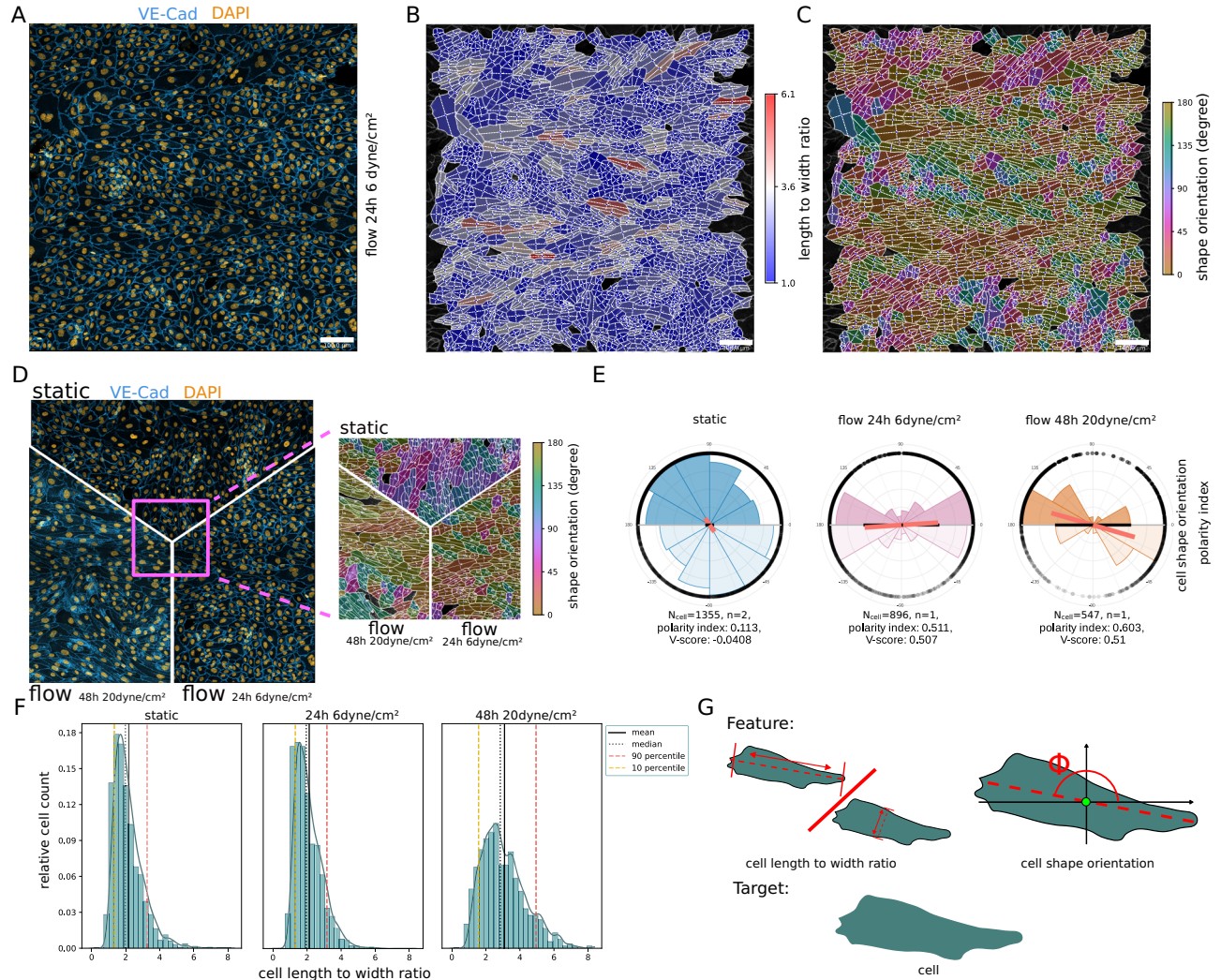

**Fig. 3 | Cell and nuclei shape orientation. A** Example input image with DAPI (yellow) and VE-cadherin (blue). **B** Cell elongation is indicated by colour with blue 'roundish' and red 'very elongated'. **C** Cell shape orientation is visualised by a circular colour scheme ranging from 0° to 180°. **D** Comparison of different flow conditions static, 6 dyne/cm² and 20 dyne/cm². **E** The orientation of cell shape is summarised in circular histograms with statistical read-outs including the polarity index (red bar) and V-score (black bar), lower hemirose plots are duplicated data points and therefore shown transparent. **F** Histogram showing the change of cell elongation with respect to the three different flow conditions. **G** Pictogram showing the calculation of the length-to-width ratio and shape orientation, which is applied to each single cell mask (target). Note, that the number of biological replicates is $n = 2$ for static, $n = 1$ for 6 dyne/cm² and $n = 1$ for 20 dyne/cm². The full set of features with target cell can be found in Supplementary Table 6. Source data are provided as a Source Data file.

the cue directional intensity ratio as a linear feature, which can loosely be described as the ratio between the mean intensities of the left-hand and right-hand cell-half of a cell perpendicular to a given cue direction. We provide a detailed description in the 'Materials and Methods' section. The ratio takes values ranging from [−1, 1] where −1 indicates a strong asymmetry against a direction of the cue, 0 without visible effect, and +1 a strong asymmetry along the direction of the cue. An example image together with a visualisation of the cue directional intensity ratio and marker polarity is shown in Fig. 4A (from left to right). A similar approach of using ratios on opposite sides of the cell was introduced by ref. 43 to determine the magnitude and angle of polarity of a given cell.

To investigate the polarity of the Notch1 signal, we compared ECs in static conditions and exposed to shear stress for a 2 h time period. The images contained a junction and nucleus channel as well as a NOTCH1 staining. The junction and nucleus channel were used for segmentation to then quantify the intracellular NOTCH1 signaling gradient.

The result of the analysis is shown in Fig. 4 bottom row. We observed a shift in the marker cue directional intensity ratio when comparing static and shear stress conditions. We observed a rather small, but consistent change in mean, indicating a small gradient of the NOTCH1 signal, with a lower concentration on the flow-facing side, Fig. 4B. The results of the marker polarity analysis showed strong asymmetry effects, see Fig. 4C, with large polarity index of 0.647 and 0.47 and V-score of 0.639 and 0.458 at 30 min and 120 min, respectively, while the mean was pointing along the direction of flow. This confirms the results from ref. 42. A circular-linear correlation analysis between the polarity index and the directional intensity ratio of the cue revealed a correlation coefficient of 0.721, see Fig. 4D, which implies a strong correlation between both measurements. Since signaling gradients in a single cell are rather small and sometimes noisy[44], ratio values are expected to be close to zero and show higher variance. However, the cue directional intensity ratio is a linear feature, easy to interpret, and can be compared between experimental setups, which is why we included it in our set of features. The marker polarity, on the

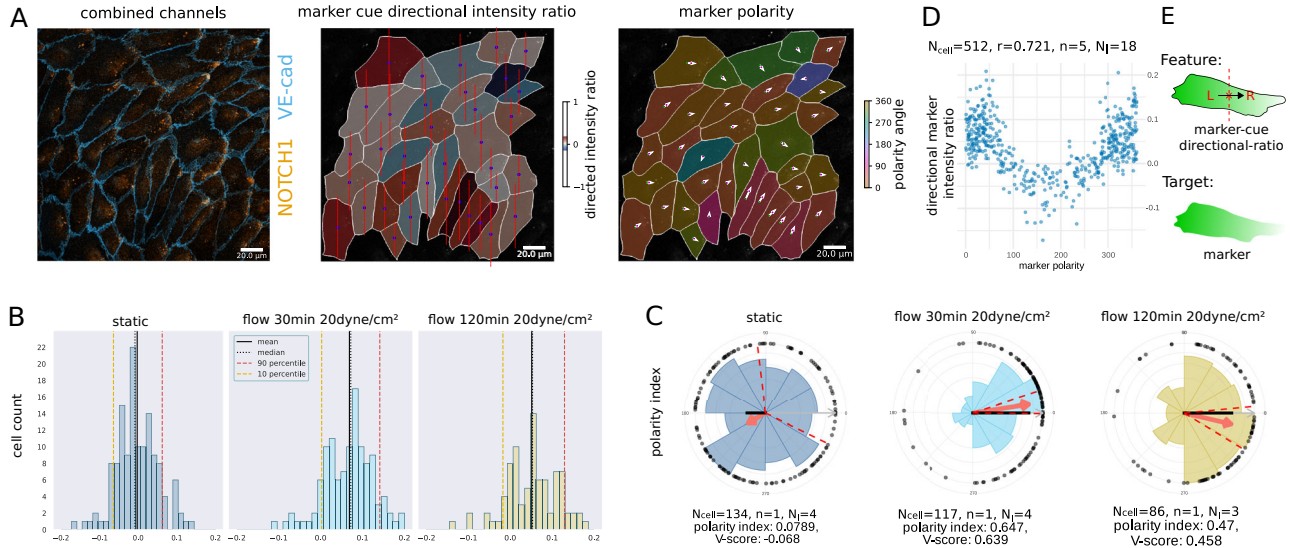

**Fig. 4 | Flow-induced NOTCH1 signaling asymmetry. A** Left shows the NOTCH1 signal (orange) combined with the junction channel (blue). Middle shows the NOTCH1 (here called marker) cue directional intensity ratio. The blue circle inside each cell marks the centroid of the cell, while the vertical dashed red lines are perpendicular to the flow direction and thus separate the cell to front and back. Right shows the marker polarity as a directional property. **B** Shows the cue directional intensity ratio of the NOTCH1 signal with respect to the flow direction before [left], after 30 min [middle], and after 2 h [right] of exposure to a shear stress level of 20 dyne/cm². **C** The circular histogram of the polarity of the NOTCH1 signal before applying a shear stress level of 20 dyne/cm² [left], after 30 min of exposure [middle], and after 2 h [right]. The red arrow points towards the collective mean direction of polarity, and its length represents the polarity index. Dashed lines indicate the 95% confidence intervals of the circular mean. Black dots indicate single-cell measurements. **D** Correlation between marker polarity and directional intensity ratio. Note that for the correlation plot, all three conditions are pooled, while the number of biological replicates for each of the flow conditions is $n = 1$. **E** Schematic representation of the feature and target for this figure. For an overview of features that can be extracted from intensity measures, see Supplementary Table 7. Source data are provided as a Source Data file.

---

contrary, is much more sensitive, but does not measure the magnitude of the gradients, but only their direction.

## Intracellular intensity patterns

To complement our investigation of signal intensity gradients, we also characterised the localisation of image-based signal intensities, which indicate the localisation of specific processes. Localisation of cellular processes in biological cells is important because it allows for precise regulation of downstream processes, such as gene expression, protein synthesis, signaling and other cellular activities. For instance, it is important whether molecules are localised at the cell membrane, where they might get activated via phosphorylation, or if they fulfill other functions through anchoring to the membrane. Localisation to the nucleus is also important for a variety of cellular processes, including gene expression or RNA processing.

To quantify the signal in the different subcellular compartments, we computed the total amount and concentration of signal intensity, in the nucleus, the cytosol (without the nucleus) and the membrane nucleus, see Fig. 5A. We demonstrate the capabilities of the Polarity-JaM pipeline, by quantifying the intensity ratio of Krüppel-like factor 4 (KLF4) in the nucleus with respect to the cytosol. KLF4 is a transcription factor that is known to be upregulated via exposure to laminar shear stress[45,46]. We calculated the intensity of KLF4 in the nucleus and cytosol for static, after 4 h and 16 h of 6 dyne/cm² flow. We found a significant increase after 4 h of flow exposure in nuclei localisation compared to control and a slow decrease at 16 h compared to 4 h Fig. 5B, C. For statistical comparison, we have used the DABEST method[47], see Fig. 5C.

## Junction morphology

Cell-cell junctions underpin any architecture and organisation of tissue. They vary in different tissues, organs, and cell types and need to be dynamically remodelled in development, homeostasis, and diseases.

For example, EC-cell junctions must provide stability and prevent leakage while also allowing dynamic cellular rearrangements during sprouting, anastomosis, and lumen formation[10]. The organisation and topology of junctions and inversely the organisation and topology of ECs contain a wealth of biological information. By analysing adjacency patterns in ECs, organisational patterns that are associated with tissue phenotypes can be uncovered. There are vast differences in endothelial arrangement between different tissues and organs[48].

We are using the cell-cell contact features from an already published tool JunctionMapper to decipher cell-cell junction-related phenotypes[11]. Note, that this tool is not adapted to studying EC in organs or in 3D tubular structures, which will be in the scope of future studies. The normalised junction features suggested in JunctionMapper allow one to quantify images of different resolution, cell type, and modalities. In our tool, the analysis is automatically performed in a region that is defined by the cell outlines, which we obtain from the instance segmentation using the proposed deep learning frameworks. This outline is dilated by a user-defined thickness. There are no more parameters necessary to define. The result can be seen in Supplementary Fig. 1C. The resulting area of the dilated outline is the interface area, which is computed for each single cell. We then derive the characteristic from a junction label, for example, VE-cadherin staining in the case of ECs, see Fig. 6A. The junction protein area results from Otsu thresholding in that region, see Fig. 6B and Supplementary Fig. 1C. Using these readouts, we can compute three features: (1) interface occupancy by computing the junction protein area over the interface area, (2) the intensity per interface area by computing the average intensity in the interface area, and (3) cluster density by the average intensity in the junction protein area. We find a unique signature of the three junction features in ECs after flow stimulation, see Fig. 6C, demonstrating the effectiveness of our method. In static condition, the cell-cell junctions are very heterogeneous, with some cells having thick junctions and high VE-cadherin intensity, while

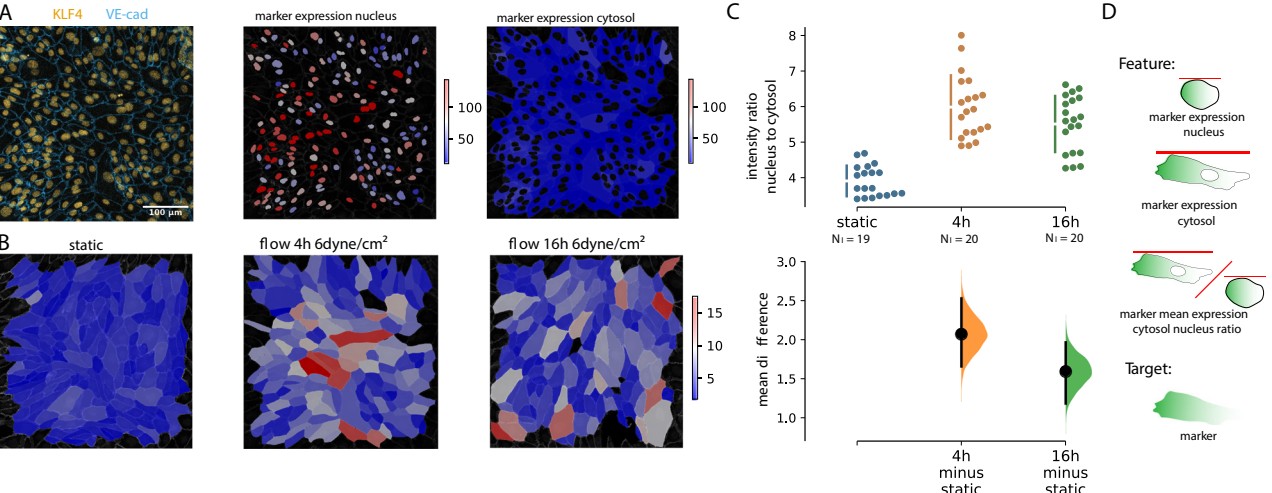

**Fig. 5 | Localised single-cell fluorescence intensity quantification.**
**A** Quantification of the intensity of KLF4 antibody staining. Overlay of KLF4 (Orange) and VE-Cad (blue) channel (Contrast enhanced for optimal view)[left], KLF4 intensity in the nuclei [middle] and in the cytosol without nucleus [right]. **B** Intensity ratio of the KLF4 reporter in the nucleus and in the cytosol was colour coded for each single cell. **C** Comparison of the change in intensity ratios using the DABEST method shows a difference between static conditions and endothelial cells

exposed to flow after 4 h and 16 h. The total number of cells analyzed is $N_{cell} = 2860$, $N_{cell} = 3467$, and $N_{cell} = 3172$ for static, 4 h, and 16 h, respectively. Each point indicates an image of $n = 2$ biological replicates for each condition. Black bars indicate 95% confidence intervals of the mean difference. **D** Schematic representation of the features and target for this figure. For an overview of features that can be extracted from intensity measures, see Supplementary Table 7. Source data are provided as a Source Data file.

others have low signal intensity and low occupancy. Intensity and occupancy become more homogeneous after exposure to flow. At 6 dyne/cm² the total intensity per interface area increases as well as the interface occupancy. At 20 dyne/cm², however, the junctions become thinner, resulting in lower interface occupancy, while the intensity per interface area remains almost the same compared to static. At the same time, the intensity within the junction increases, resulting in higher values of cluster density. For the entire junction analysis workflow, only one additional parameter needs to be specified, namely the width of the automatically generated outlines that serve as regions of interest for cell-cell contacts. In summary, Polarity-JaM offers the possibility to fully automate the essential parts of the JunctionMapper workflow by setting a single additional parameter.

## Reproducability, replicability and interoperability
Iterative acquisition of images and various experimental settings sometimes require complex folder structures and naming schemes to organise data, leaving the researcher with the problem of data structure and replicability of their analysis. To help with both tasks, the Polarity-JaM pipeline has three execution scenarios: (a) single image, (b) image stack, and (c) complex folder structure—for the latter option see Supplementary Table 9. Furthermore, a comprehensive logging output is provided, as well as a standardised input structure in yml format. For a list of configurable parameters we refer to Supplementary Table 10. The generated outputs follow a naming scheme. The extracted collective and single cell features are stored in a csv file. The results of the statistical analysis from the app can be downloaded in various formats, including pdf and svg. For different categories or conditions, the Polarity-JaM app uses several qualitative colour schemes that are colour-blind friendly, the infrastructure follows a similar principle as PlotTwist[49], which was designed for time series analysis. Metadata and log information are also saved in a human-readable format on disk. Polarity-JaM uses well-established non-proprietary formats (such as csv, yml, tiff, svg) to aid interoperability, following a recommendation in ref. 50. All statistical analysis for circular features shown in this study and more can be performed in the App. Our tool can be combined with other tools such as Griottes[21], polarity features can be mapped on spatial network graphs and their

relation can be explored using the same segmentation, see Supplementary Tables 12 and 13.

Exploitative image analysis requires interactivity to quality check each analysis step. Hence, Polarity-JaM is designed with a simple Python API that is optimised for usage within a Jupyter Notebook[51]. We provide several examples in our documentation on how to perform such an analysis. An overview of the entire Polarity-JaM software suite is depicted in Fig. 7. We additionally equip Polarity-JaM with a Napari[52] plugin with a graphical user interface to enable direct feedback on segmentation and features. Finally, Polarity-JaM is available via Python Package Index (PyPI). Taken together, we are committed to the principles of FAIR research[53].

## Discussion
Our image data processing and analysis workflow can be used to simultaneously compute features of cell polarity, including organelle localisation, cell shape, and signalling gradients, allowing single-cell and collective high-content endothelial phenotyping. Circular statistics can be performed interactively via a web application. We also provide an informative graphical design for directional and axial data. We recommend the use of the signed PI when a polarisation direction is expected on the basis of an external cue, as is the case in most endothelial flow assays.

With the focus of the Polarity-JaM toolbox on a diverse feature set and replicability and interactivity, we provide means for answering various biologically motivated experimental questions and for the extraction of aspects that are otherwise easy to overlook. For instance, collective orientation and cell size have been inversely correlated with EC senescence. Older ECs tend to be larger and share a less pronounced direction of orientation[54]. The orientation as a collective phenotype can be correlated with processes such as flow and senescence. Multivariate analysis, for example, including size in the image analysis, could perhaps differentiate between endothelial phenotypes by flow or those by cellular senescence.

This investigation has been exemplified for ECs, but can also be applied to other cell types such as cardiomyocytes[55], epithelial cell tissues[20] and potentially other cell types. The image modality is not restricted to fluorescence microscopy but can be also applied to phase

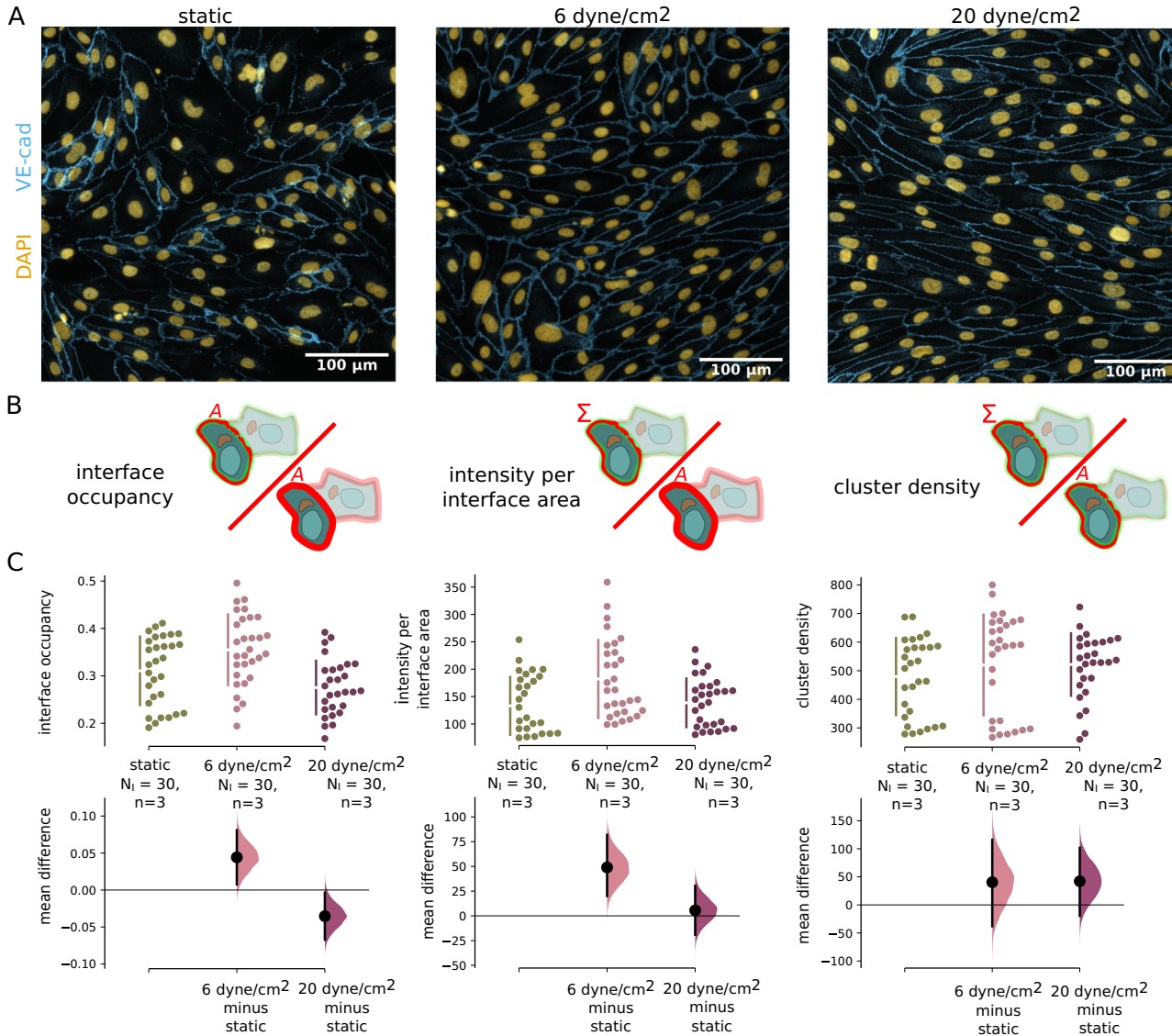

**Fig. 6 | Junction morphology quantification. A** Examples of endothelial cells in static condition, exposed to 6 dyne/cm² and 20 dyne/cm². VE-cadherin staining is shown in blue, DAPI in yellow. **B** Pictograms of quantities extracted from the images. **C** Statistical comparison of three normalised morphological junction features interface occupancy [left], intensity per interface area [middle] and cluster density [right] demonstrate a significant change in junction dynamics after exposure to flow. Black bars indicate 95% confidence intervals of the mean difference. An overview of features that can be extracted from cell-cell junctions is shown in Supplementary Table 8. Source data are provided as a Source Data file.

contrast or other—here the only restriction is that the image can be decomposed into masks of single cells. The current version of Polarity-JaM integrates different segmentation models, including Cellpose[9], microSAM[8] a fine-tuned model based on SAM[7], and DeepCell[6]. There is a vivid community around all these segmentation algorithms[8,9]; therefore, we provide an interface to these models, which can also be adapted by the user. This will help maintain this software and ensure long-term use. Future development needs to address better segmentation of subcellular structures, including cell-cell junctions, cytoskeleton, and FA sites, using deep learning methods.

The quantification of junctional morphology is based on the features suggested in ref. 11 including junction occupancy, cluster density, and intensity per interface area. While these features provide good indicators for junctional changes and adaptations, they may not be exhaustive referring to the manual morphological classification of adherens junctions, which is frequently done in five common categories: straight junctions, thick junctions, thick to reticular junctions, reticular junctions, and fingers[56]. To automate the translation into this

classification, further work is needed on junction segmentation, as well as an advanced classifier using manual training data, which was not ready at the time of this publication but will be addressed in the future.

Future challenges involve tissue and organoid image data in 3D space, which introduces more challenges in algorithmic development including robust segmentation (mainly due to the lack of training data), anisotropy in image acquisition, and the size of the image data. Also, efficient extraction of cell and nuclei features, which are by default not included in common packages, need to be developed. Multiplex imaging will stimulate further developments as this image data modality dramatically increases the information content and therefore challenges meaningful feature extraction and comprehensive spatial and circular statistical approaches[57].

The focus of this pipeline is on static images. However, the pipeline could also be applied to a series of images, and feature extraction would be performed for each frame. The extracted data can be stitched together by label identification[58]. Computational models can be informed by a wealth of quantitative data through our approach,

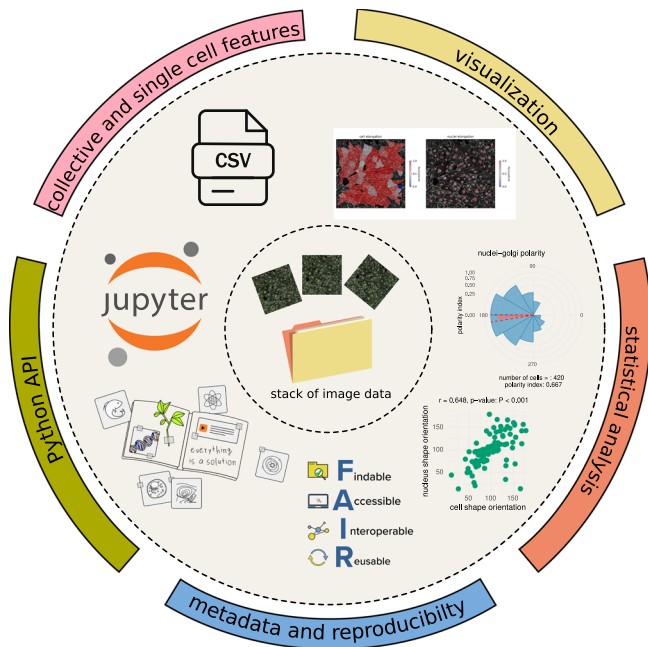

**Fig. 7 | Workflow for large image data stacks with automated feature extraction and quality control.** The workflow enables processing large image data stacks for automated feature extraction. The Python API can be used to facilitate quality control with the help of a Jupyter Notebook. An overview of the Python application interface (API) is provided in Supplementary Fig. 7. The results in comma-separated value (CSV) table format can be statistically analyzed using a graphical interface.

including vertex models[35], but also the cellular Potts model[34,59] or agent-based models[60,61]. The spatial context can be further explored using tools such as Griottes[21] and the circular version of Moran's I[62] to extract collective phenotypes. Mechanistically, this will also help to predict different states of tissues and dynamics from static biomedical images[36,63,64], which is an interesting avenue for future research with a wide range of applications.

## Methods

### Experimental setup

**Cell culture and shear stress assays.** Commercially available human umbilical venous ECs (HUVECs, mixed donors of both sexes, Promo-Cell: C-12203) were cultured and used at passages 2–4 at 37 °C and 5% $CO_2$ humid incubator, in endothelial growth medium containing growth supplement kit (EGM2, Lonza) for optimal cell growth. For passaging and fluid shear stress (FSS) assays, cells were washed once in sterile PBS, followed by a 5 min incubation in Trypsin at 37 °C and 5% $CO_2$, then neutralised with FBS and EGM2. Cells were centrifuged for 5 min at 480 × $g$ and counted. For FSS assays, cells were seeded in 0.4 ibiTreat Luer flow slides (Ibidi) coated with 0.2% gelatin at a cell concentration of 2 million cells per ml. One hundred microlitre of cell suspension were added to each slide and incubated overnight at 37 °C and 5% $CO_2$. The following day, the slides were connected to red perfusion sets assembled onto perfusion units (Ibidi) and connected to a pump (Ibidi). Laminar shear stress was applied at 6 dyne/cm² or 20 dyne/cm² for 4, 16, or 24 h inside a 37 °C and 5% $CO_2$ humidity incubator. Static controls were kept in the same incubator for the duration of the experiment.

**Immunofluorescence.** At the end of flow application, slides were disconnected from perfusion units and immediately fixated in 4% PFA for 10 min, then washed three times in PBS. Slides were then blocked and permeabilized in BB for 3 min, followed by 1 h of incubation with

primary antibodies at room temperature. Cells were washed 3 times with PBS, then incubated 1 h with secondary antibodies at room temperature, followed by another triple washing step, 5 min incubation with DAPI, and finally mounted with a Mowiol + Dabco mixture in a 9:1 ratio. The primary and secondary antibodies used can be found in Supplementary Table 11.

**Confocal image acquisition.** FSS immunostained slides were imaged on a confocal microscope (Carl Zeiss, LSM 980) using a Plan-Apochromat 20×/0.8 NA Ph2 air objective and 63x/1.4 NA oil objective. For each sample, random positions throughout the flow slide were selected and Z-stacks were acquired covering the entire depth of the monolayer. Slides were imaged with a two-channel setup, with channel one using the 488 and 633 lasers and channel two using the 405 and 561 lasers. Pinhole size was set to 1AU for both channels. ZEN version 3.4.91.00000 was used for image acquisition, Fiji[65] was used for max projection and export to tiff file format. Raw data and tiffs were stored and processed on the internal OMERO[66] server of the Max Delbrück Center for Molecular Medicine.

### Image analysis

**Segmentation.** To isolate individual cells in a microscopic image, a process also known as instance segmentation, we used Cellpose, a deep neural network algorithm. Accurate instance segmentation can be created with pre-trained models it is provided with. These models can generalise well across both cell type and image modalities. For our analysis, we used the model 'cyto' for cell and 'nuclei' for nuclei instance segmentation. For Golgi segmentation, Otsu-thresholding was performed. Subsequently, the segmentation mask was used to get the corresponding Golgi instance label. The performance of instance segmentation algorithms can vary for different modalities. Downstream analysis of features describing individual cells and their relationship with each other strongly depend on the quality of these segmentations. At this point Polarity-JaM offers three segmentation algorithms that the user can choose from: Cellpose, DeepCell, and microSAM which are individually configurable (see Supplementary Table 14). Additional segmentation algorithms are realised with the help of Album[67], a decentralised distribution platform where solutions (in this case implementations of segmentation algorithms) are distributed with their execution environment and can be used without additional overhead for the user.

**Single-cell and organelle features.** Common features are available within the scikit-image package[68]. We extend the available measurements by various features. For a complete list of all features, see Supplementary Note 2.

Most features require central image moments[69] that can be calculated from the raw moments

$$m_{i,j} = \sum_x \sum_y x^i \cdot y^j \cdot I(x,y), \tag{5}$$

with $i, j = 0, 1...$ are exponents, $x,y$ the pixel coordinates, where $I(x, y)$ refers to the image intensity at position $x,y$. Generally, the centre of mass of a grey scale image (e.g. a channel) is now given by

$$M = (\bar{x}, \bar{y}) = \left( \frac{m_{1,0}}{m_{0,0}}, \frac{m_{0,1}}{m_{0,0}} \right). \tag{6}$$

The central moments are then

$$\mu_{i,j} = \sum_x \sum_y (x - \bar{x})^i * (y - \bar{y})^j * I(x,y). \tag{7}$$

**Shape orientation.** With the central moments, we compute the orientation

$$\phi = \sigma \frac{1}{2} \operatorname{atan2}(2\mu_{1,1}, \mu_{0,2} - \mu_{2,0}) \quad (\bmod \pi), \qquad (8)$$

which describes the angle of the major axis of the object (e.g. nuclei or cell shape) with the $x$-axis in the interval $[0, \pi]$ in radians or 0 to 180°. Note that $\sigma = \pm 1$ depends on the choice of coordinate system of the biomedical image, e.g. $\sigma = +1$ is used if the $y$-axis is from bottom to top, while $\sigma = -1$ is used if the $y$-axis is from top to bottom. Various features can be defined with the orientation, such as the cell shape orientation or the nucleus orientation in case the nucleus channel is provided.

**Directed front-rear polarity features.** Directed or front-rear polarity features such as nuclei-Golgi polarity, nucleus displacement, or marker polarity are generally defined as angles $\alpha$ by

$$\alpha = \sigma \cdot \operatorname{atan2}(\bar{y}_f - \bar{y}_r, \bar{x}_f - \bar{x}_r) \quad (\bmod 2\pi), \qquad (9)$$

where index $r$ indicates moments that are calculated on a reference or 'rear' target, and $f$ moments that are calculated on the 'front' facing target. In this way we can define directed front-rear polarity based on two targets. In the case of nuclei-Golgi polarity, the target is the Golgi channel and the reference is the nuclei channel. In the case of nuclei displacement or marker polarity, the centre of mass of the cell is the reference and the 'front' facing target is the nucleus or the intensity weighted centre of mass, respectively. The values of the directed polarity features take values in $[0, 2\pi]$ in radians, which corresponds to 0° to 360°, in this study.

**Signaling gradient quantification.** We define the cue directional intensity ratio for a cell as

$$s_r = \frac{(1 - 2I(x,y)A_l)}{I(x,y)A_{cell}}, \qquad (10)$$

where $A_l$ is the area of the left cell half perpendicular along a given a cue direction $\alpha_p$ and area of the cell $A_{cell}$. Mathematically, the area $A_l$ is described as

$$A_l = A_u \cap A_{cell} \qquad (11)$$

with

$$A_u = \left\{ \bar{v} = (x,y) \middle| \left( \bar{v} - \begin{pmatrix} \bar{x} \\ \bar{y} \end{pmatrix} \right) \cdot \bar{v}_c < 0 \right\}, \qquad (12)$$

and $\bar{v}_c = (\cos(\alpha_p), \sin(\alpha_p))$, where the polar direction or expected cue direction $\alpha_p$ is given in radians and $\bar{x}$ and $\bar{y}$ are calculated over the cell mask.

**Statistics and reproducibility.** We used the notation $N_{cell}$, $N_{nuc}$ for numbers of analyzed cells or nuclei, respectively. Furthermore, $N_I$ indicates the number of images analysed and $n$ the number of biological replicates. No statistical method was used to predetermine sample size. No data were excluded from the analyses. The experiments were not randomised and the investigators were not blinded to allocation during experiments and outcome assessment. All statistical methods were implemented in R version 4.4.1, the statistical analysis from the CircStats package (version 0.2-6)[28] for directional and axial data. We also used the DABEST ('data analysis with bootstrap-coupled estimation', version v2024.03.29) method[47].

## Reporting summary

Further information on research design is available in the Nature Portfolio Reporting Summary linked to this article.

## Data availability

The image data generated in this study have been deposited in the BioImage Archive database under accession code S-BIAD1540 https://www.ebi.ac.uk/biostudies/bioimages/studies/S-BIAD1540. The extracted numerical feature data generated in this study are provided in the Source Data file. Source data are provided with this paper.

## Code availability

The pipeline was developed in Python and is available through PyPI. The code for the R-shiny application was written in R and Rstudio (https://www.rstudio.com). The Polarity-JaM app can be used online via (www.polarityjam.com) without the installation procedure. Both the Polarity-JaM pipeline and the app can also be used offline. Additionally, both can be installed and used separately through album[67], a framework for scientific data processing with software solutions of heterogeneous tools. The link can be found https://album-app.gitlab.io/catalogs/helmholtz-imaging/de.mdc-berlin/polarityjam. The code for pipeline, app, and Napari plugin is published under the MIT licence and is available through GitHub (https://github.com/polarityjam). Additional documentation and information can be found at readthedocs (https://polarityjam.readthedocs.io). Issues, requests and contributions are tracked on GitHub issues. Collaboration and contributions are possible and welcome. Instructions and best practices can be found in the documentation.

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

## Acknowledgements

The authors thank the IT department of the Max Delbrück Center for Molecular Medicine, especially Frank Büttner for his help and support in deploying the R-shiny web application on the MDC server. In questions of data protection and the legal framework for the online application, we were supported by Ulrike Ohnesorge and thank her for her commitment. Furthermore, the project benefited from the Deutsche Forschungsgemeinschaft (DFG, German Research Foundation)— Project-ID 414984028—CRC 1404 FONDA. The authors also thank Christoph Karg for valuable feedback. This work was supported by the Deutsches Zentrum für Herz-Kreislaufforschung, the Bundesministerium für Bildung und Forschung, the Deutsche Forschungsgemeinschaft by the CRC1366 and grant numbers 329389797, CRC1444 and CRC1470 to H.G. This project has also received funding through a grant from the Fondation Leducq (17 CVD 03) to H.G. and part of this work was funded by HELMHOLTZ IMAGING, a platform of the Helmholtz Information & Data Science Incubator to D.S. and K.H.

## Author contributions

W.G.: conceptualisation, data curation, formal analysis, investigation, methodology, project administration, software, supervision, validation, visualisation, writing—original draft preparation, writing—review and editing; J.P.A.: conceptualisation, data curation, formal analysis, investigation, methodology, software, validation, visualisation, writing— original draft preparation, writing—review and editing; O.O.: data curation, formal analysis, investigation, validation, writing —original draft preparation, writing—review and editing; E.A.: investigation, methodology, writing—review and editing; J.K.: investigation, visualisation, writing—review and editing; D.S.: funding acquisition, resources, writing—review and editing; K.H.: conceptualisation, funding acquisition, methodology, supervision, visualisation, writing—review and editing; H.G.: conceptualisation, funding acquisition, resources, supervision, writing—review and editing.

## Funding

## Competing interests

The authors declare no competing interests.
