## [Transparent Peer Review file · Nature Communications]

Polarity-JaM: An image analysis toolbox for cell polarity, junction and morphology quantification

Corresponding Author: Dr Wolfgang Giese

Version 0:

Reviewer comments:

Reviewer #1

(Remarks to the Author)

In the article entitled, "Polarity-JaM: An image analysis toolbox for cell polarity junction and morphology quantification", the authors present and make available their Polarity Jam software. This is a very power software that will be a significant addition to the field. It will improve the way that mechanotransduction is studied and reported, and therefore has a strong chance of changing the way that we study mechanotransduction. As such, the software is a significant contribution to our field.

My only concerns regard the readability of the article, as well as some concerns about the way that the data is presented.

- 1) Both "Asymmetries in subcellular localisation of organelle" and "Shape asymmetries" sections begins with one or two paragraphs that are written more in the style of a review article than results. I understand that the authors want to indicate why the coming results/measures are of interest, but these should be shortened significantly.
- 2) Statistics are performed on individual nuclei and do not indicate the number of experiments. I understand that with such a rosette, the analysis must be individual cell/nuclei, but the number of repeats should also be indicated. Stating "N>3 for all panels" is not sufficient. This could easily be included in the text that indicates the number of nuclei, e.g. "NNUC = 94, n=3, r=0.635, P < 0.001"
- 3) In figure 2B, the first rosette shows grey and red arrows in the opposite directions and the signed polarity is negative. In the second rosette, the grey and red arrows are in the same direction and the signed polarity is still negative. I believe that second rosette is therefore mislabeled.
- 4) The alpha angle is not properly defined until the methods. I realize defining the math of alpha requires the authors to define the moments, and the shapes, and so forth. But alpha could be defined in words in the main text. A sentence such as "Alpha is the orientation of the displacement of given parameter (nucleus-golgi, nucleus-center of mass, etc.)." Or define it on figure 2A. Which brings about another point, in the main text, polarity index is defined based on nucleus-golgi, and then suddenly, other polarity index's are presented without really explaining that there are several possible alpha's. This should be explained more explicitly and earlier on, because currently it takes some effort on the part of the reader to figure out these aspects of Polarity Jam.
- 5) How can static conditions have a signed polarity index? I realize that the authors are using the direction of flow used on the other slides, but this is not an appropriate thing to do. V should be undefined for static conditions.
- 6) I do not understand the sentence "Here any angle α_i is identified with its opposite $\alpha_i + 180$ thus we do not distinguish the front and back of the cell (or nucleus)." More specifically, how can an angle be "identified with its opposite". What does mean to define an angle? And how was defining it with its opposite achieved?
- 7) I am generally not clear what Figure 3 is showing. The section is titled "Shape asymmetries" and yet all the measures are about orientation, not asymmetry. Is it asymmetry of one cell compared to another? Or asymmetry within a given cell? A clearer title than "shape assymetries" could help readers understand which is meant. At first, I assume that the authors meant the earlier, since the rosette shows cells at both extremes and the region in the box has many cells that are round and many

that are elongated. But no measure for this asymmetry is provided and so it is an observation of cell behavior within the author's data, and it is not something that the software can "do", i.e. a measure of asymmetry. This article should be limited to what the software can do. But then, the last paragraph of this section made me doubt my assumption that the authors referred to inter-cell asymmetry and rather were talking about intracellular asymmetry (because the text begins to refer to fitting cells to ellipses and so forth). But again, no measure is provided. In fact, the conclusion seems to be that such fitting is inappropriate. I would completely remove figure 3. Alternatively, a new aspect of the software should be introduced.

8) In figure 5B, there are some cells in the middle of images that appear not to be included (i.e. they are black whereas all other cells are between shades of blue or red). Why were they excluded? Did they lack a nucleus? Some could be gaps in confluence, but many look very cell-shaped. In fact, figure 5A has a gap that definitely has a nucleus, but I will admit that the nucleus looks very large. The authors should better define the exclusion criteria used.

9) Figure 6B defines the 3 junction measures, but it is not clear how these measures are defined. How is the interface defined? How is the fragmented junction area defined? I thought this would be present in S1 but it is not. It is measured in S2, but I still don't understand how these regions are defined.

10) Supplementary table S5 is unnecessary. If you take out the sentence "The categories are shown in table S5.", then each category is already nicely defined in the main text of appendix 3. It also reads better because you are not distracted by flipping to table S5.

Minor

Figure 1 – flow direction should be indicated in figure not just in legend
There is a typo for "nuclei".

For figures 3A/5A, the stains used are not indicated in the legend. Figure 6 indicates what is blue but not what is yellow.

What level of shear stress was applied in figure 4, only the duration is indicated.

In the methods, under "Nucleus and organelle displacement", the authors state "The displacement orientation from the nucleus to the centre of mass of the cell can be defined as $\alpha = \sigma \cdot \arctan 2(\bar{y}_t - \bar{y}_r, \bar{x}_t - \bar{x}_r)$ ", but what is defined is then actually an alpha for any two measures (organelle-nucleus, nucleus-cell, etc). That first sentence indicates that alpha is only for nucleus-centre of mass. The part of the sentence "from the nucleus to the centre of mass of the cell" is incorrect (or specific to one condition), I believe.

Unless required by the journal, the supplemental tables should come in line with the text. I was reading the methods, which sent me to appendix 3, which then sent me to supp table 5. I did "ctrl-f" and every supplemental table is called only once in the pdf. Therefore, they could easily be placed in line with the text.

The main text refers to Appendix 3, but there are no appendices, only "Supplementary notes". If the supplementary note is the appendix, please use one term.

μ is both the angle for the flow and the central moment. A different symbol should be used for the flow angle.

In the methods, the title of the section is "Nucleus and organelle displacement" but it should be "Nucleus and organelle displacement orientation". An angle not a displacement is defined.

(Remarks on code availability)

Reviewer #2

(Remarks to the Author)

The manuscript describes a new software tool "polarity-jam" for analyzing directional cell features, e.g. of endothelial cells in blood vessels affected by flow. This software integrates existing cell segmentation tools with a python based pipeline for feature extraction and a web-based tool for feature analysis.

This work could provide a valuable software package for cell feature analysis, as it integrates segmentation with feature analysis for an important application in a user-friendly manner (with a few caveats, see comments on below). The availability of off-the-shelf and generalizable cell segmentation algorithms is increasing the need for such analysis tools that enable standardized downstream analysis for specific applications. As such I believe that this work is suitable for publication in Nature Comms. However, I believe that a few changes are necessary to improve the clarity of presentation and the documentation of the software.

Regarding the manuscript: The manuscript is overall well written and the graphics represent the results well and are easy to understand. The methodology is clearly described and contains enough details for reproducibility. The conclusions drawn are warranted given the presented data.

However, the manuscript does not provide a fully clear overview of the features provided by polarity-jam and which features

match based to a user's problem. The first 5 result subsections, "Asymmetris in subcellular localisation of organelles", "Shape asymmetries", "Quantification of intracellular signalling gradients", "Localized marker expression of KLF4" and "Junction morphology" give examples of different analysis that can be performed based on the cell features implemented in polarity-jam, but an easy overview of the different features that are available and their applicability is missing. I suggest to first give a high-level description of the different feature categories in an introductory paragraph of the results and also add a new sub-figure to Fig. 1 that visualizes this overview. Please also indicate the differences between these features clearly, e.g. I am currently unsure what the difference between "nuclei-organelle polarity" and "fluorescent marker polarity and intensity" is, since the organelle / Golgi analysis is also based on a fluorescent marker. Each result subsection should then indicate which of the feature categorie(s) is used.

Regarding the code and documentation: Overall, I found the code easy to install and to use. However, the documentation of the code is still fairly shallow. In particular, I did not find any documentation for the napari plugin; the github repo references the wiki where no further information on the plugin can be found. I think that the napari plugin is crucial for visualization purposes to verify the segmentation results in an easy manner, since the reliability of the analysis hinges on a correct segmentation. It would also help computationally less versed users to make use of the plugin. Furthermore, I am also missing a section on "Which feature should I use for my analysis problem" from the wiki.

<https://polarityjam.readthedocs.io/en/latest/Features.html> just lists the available features without much further explanation. While I understand that it is not possible to exhaustively address all possible use-cases it would be very helpful to have some examples and guides that explain which features to use given a certain imaging condition. E.g. "I have a cytosol and nuclear stain with additional golgi fluorescent marker. Which features can I use for analyzing reaction of my cells to different directed flow conditions." A few of these examples would clearly help users of the tool. In this context a walk-through tutorial, either as a video or a page with screenshots, that presents one or more example analyses from start to finish would also help. Finally, I think it would make sense to add support for custom, already computed, segmentation results for cases where the in-built methods of polarity-jam are not a good fit and users already have a custom segmentation solution.

In summary: I think this is a valuable and technical sound contribution. Before acceptance I recommend the following improvements (concise summary of the points I made above):

- Provide a concise overview and distinctions of the implemented features at the beginning of Results and in Fig. 1.
- Add documentation for the napari plugin.
- Add documentation that explains the different features with a focus on which features are applicable for a given experimental condition and analysis problem.
- Add tutorials that show a walk-through of using the tool (either screenshots or video, can be combined with the previous point).
- Add support for external segmentations. (This point is not as important as the others, since the tool is still useful without. It may also be possible through the Python API already, which I have not reviewed in much detail. If that is the case maybe just add a short section on this in the doc)

Details: Figure 1 is missing "A", "B" as well as the black arrow that should indicate the flow direction according to the caption.

(Remarks on code availability)

See main comments.

Version 1:

Reviewer comments:

Reviewer #1

(Remarks to the Author)

In the revised version, the authors have made significant improvements that help with the accuracy and readability of the article as well as adding several features that will help other labs get up and running with the application.

I still have two very minor points, but important. Because, within the introduction paragraphs there is one statement that is technically incorrect and a second statement that is absolutely incorrect.

1) The authors refer to "fluid shear stress". It is actually a wall shear stress. Shear stress is $\mu \cdot dv/dr$. dv/dr has a value at all values of the radius, even in the fluid. But endothelial cells sense the shear stress at $\mu \cdot dv/dr$ at $r=R$. And this is the wall shear stress. See figure 2 of https://www.homepages.ucl.ac.uk/~uceseug/Fluids2/Notes_Viscosity.pdf. It defines both the shear stress at a random y (which is a fluid shear stress) and at $y=0$ (which is the wall shear stress). I realise this is commonly called "fluid shear stress" by biologist, but that's actually incorrect. The whole discussion can be avoided by simply saying "ECs are sensitive to shear stress when blood flow passes it runs through blood vessels". I would therefore advise the authors to remove the word "fluid".

2) "Collective cell orientation was also correlated with both types of shear flow, pulsatile and laminar, but pulsatile flow does not organise orientation direction as strongly as laminar flow".

This sentence does not make any sense. Pulsatile flow is laminar. I think the authors mean steady flow versus pulsatile. I realise this wording existed in the author's first submission, but I missed it. The references for this statement also do not

make sense. Ref 17 is about valvular endothelial cells (Nandini et al). Valvular endothelial cells are special, they align perpendicular to flow rather than parallel and show other unique behaviours. But Ref 17 is also a very bad paper. They use a soft matrix when a valve is cartilage and actually quite stiff. They never identify the source of valvular endothelial cells (especially concerning since their "endothelial cells" express aSMA). They never define their pulsatile wave form or the frequency of the paper pulsation. Plus, they show the opposite of the author's sentence. They show greater loss of circularity, greater increase in aspect ratio at 48h pulsatile than 48h "laminar". The second reference for this statement (Ref 18, Vion et al.) does not look at pulsatile flow. Therefore, two things are needed here. First, it must say steady versus pulsatile, not laminar versus pulsatile. And second, other references are needed. But the authors will find that the vast majority of papers find that pulsatility does the same thing as steady flow, just faster and more pronounced. Though perhaps the authors meant "disturbed" versus "laminar"?

And there are two typos:

Alignment on p7 should be alignment.

proteing should be protein

(Remarks on code availability)

Reviewer #2

(Remarks to the Author)

The revision has addressed (almost) all the points I have raised in my initial review:

- Figure 1 and the introduction now provide a much better overview of the available features and explains their differences.
- The documentation of the tool has improved a lot, including documentation of the napari plugin with a video.
- Several other minor points were addressed.
- The only point not fully addressed are more concrete use-cases / examples in the documentation, which could be quite helpful for potential users. This can however be easily added post publication, also based on experience with users of the software.

I thus recommend to accept the manuscript as is.

(Remarks on code availability)

The code is now well documented.

Polarity-JaM - Response to Reviewers

Reviewer #1 (Remarks to the Author):

In the article entitled, "Polarity-JaM: An image analysis toolbox for cell polarity junction and morphology quantification", the authors present and make available their Polarity Jam software. This is a very power software that will be a significant addition to the field. It will improve the way that mechanotransduction is studied and reported, and therefore has a strong chance of changing the way that we study mechanotransduction. As such, the software is a significant contribution to our field.

Response: We would like to thank you for this positive assessment and are very delighted that you consider this software and manuscript a valuable contribution.

My only concerns regard the readability of the article, as well as some concerns about the way that the data is presented.

1) Both "Asymmetries in subcellular localisation of organelle" and "Shape asymmetries" sections begins with one or two paragraphs that are written more in the style of a review article than results. I understand that the authors want to indicate why the coming results/measures are of interest, but these should be shortened significantly.

Response: We have significantly shortened both sections to avoid unnecessary lengthy explanations of the biological background and instead refer to reviews or studies elsewhere.

2) Statistics are performed on individual nuclei and do not indicate the number of experiments. I understand that with such a rosette, the analysis must be individual cell/nuclei, but the number of repeats should also be indicated. Stating "N>3 for all panels" is not sufficient. This could easily be included in the text that indicates the number of nuclei, e.g. "NNUC = 94, n=3, r=0.635, P < 0.001"

Response: We have adapted our figures to explicitly include the number of biological replicates, number of cells and, where appropriate, images in the figure (not just the caption) to make this clear and avoid confusion. We have used the notation N_{cell} , N_{nuc} for the number of cells and nuclei analysed, respectively. Furthermore, N_i indicates the number of images analysed and n the number of biological replicates. We added an explanation of this notation to the methods in the "statistics" section. Furthermore, we also changed the recommended structure of the key_file and folders (Supplementary Table S9) to indicate the number of replicates, directly.

3) In figure 2B, the first rosette shows grey and red arrows in the opposite directions and the signed polarity is negative. In the second rosette, the grey and red arrows are in the same direction and the signed polarity is still negative. I believe that second rosette is therefore mislabeled.

Response: Thank you very much for this attentive reading. We changed this important detail.

4) The alpha angle is not properly defined until the methods. I realize defining the math of alpha requires the authors to define the moments, and the shapes, and so forth. But alpha could be defined in words in the main text. A sentence such as "Alpha is the orientation of the displacement of given parameter (nucleus-golgi, nucleus-center of mass, etc.)." Or define it on figure 2A. Which brings about another point, in the main text, polarity index is defined based on nucleus-golgi, and then suddenly, other polarity index's are presented without really explaining that there are several possible alpha's. This should be explained more explicitly and earlier on, because currently it takes some effort on the part of the reader to figure out these aspects of Polarity Jam.

Response: We thank the reviewer for bringing this ambiguity to our attention. We have included a sketch showing the α in Figure 2, and added a sentence directly after the introduction of α : "Note that α in this example is the orientation of the displacement from nuclei to Golgi, but can be a placeholder for any given directed 'front-rear' polarity feature, including nuclei displacement with respect to the cell centroid and others, see Table S1." We have furthermore included a detailed feature table with pictograms to help the reader understand this and other features more easily (see Supplementary Table S2). Note that a feature table including an animation that explains how this angle is computed can also be found in our revised online documentation at <https://polarityjam.readthedocs.io/en/latest/Features.html>.

5) How can static conditions have a signed polarity index? I realize that the authors are using the direction of flow used on the other slides, but this is not an appropriate thing to do. V should be undefined for static conditions.

Response: You may have a signed polarity index for static conditions because we are testing for a given polar direction. For example, if we test for left to right polarisation in a static condition and we consistently get a large V-score, this may indicate that there is an issue with the experimental device or setup. So we can still assume a direction and test for an effect (see V-test). We therefore added an explanation in the results section: "Note that the polar direction provides a reference for comparing conditions, therefore we also calculate the V-score for the static condition, even though there is no flow."

6) I do not understand the sentence "Here any angle α_i is identified with its opposite $\alpha_i + 180$ thus we do not distinguish the front and back of the cell (or nucleus)." More specifically, how can an angle be "identified with its opposite". What does mean to define an angle? And how was defining it with its opposite achieved?

Response: Thank you for pointing this out. We removed this sentence and revised our explanation. "It is important to note that all axial orientation measurements have a periodicity of 180 degrees and are therefore repeated every 180 degrees in the circular histograms."

Furthermore, we have renamed α in the results "Shape orientation and morphology" to ϕ as this is now consistent with our method section "shape orientation". This is important as α from the first result section "Asymmetries in subcellular localisation of organelles" is a placeholder for directed (front-rear) polarity features ranging from 0 - 360 degrees (for instance nuclei-Golgi polarity).

In contrast, ϕ is an axial feature and has a periodicity of 180 degrees (for instance cell shape or nucleus orientation).

An explanatory animation can be found in the "Polarity" section of the online documentation at <https://polarityjam.readthedocs.io/en/latest/Features.html#type-polarity>, and a descriptive pictogram in Figure 3.

To further clarify the distinction, we have also revised the axial statistical plots (see Figure 3). Since axial data points repeat every 180 degrees, the resulting duplicated data points in the full circular histogram are now transparent by default (although the user can change the transparency to their preference).

Note: The terms directed and axial are taken from the standard book "Topics in Circular Statistics" (by S. Rao Jammalamadaka, Ambar Sengupta, Ashis Sengupta), but are often used ambiguously in the scientific literature, therefore we think that this section is very important.

We thank the reviewer again for raising this clarification and hope that these explanations and additional infographics will clarify the distinction for our readers and software users.

7) I am generally not clear what Figure 3 is showing. The section is titled "Shape asymmetries" and yet all the measures are about orientation, not asymmetry. Is it asymmetry of one cell compared to another? Or asymmetry within a given cell? A clearer title than "shape assymetries" could help readers understand which is meant. At first, I assume that the authors meant the earlier, since the rosette shows cells at both extremes

and the region in the box has many cells that are round and many that are elongated. But no measure for this asymmetry is provided and so it is an observation of cell behavior within the author's data, and it is not something that the software can "do", i.e. a measure of asymmetry. This article should be limited to what the software can do. But then, the last paragraph of this section made me doubt my assumption that the authors referred to inter-cell asymmetry and rather were talking about intracellular asymmetry (because the text begins to refer to fitting cells to ellipses and so forth). But again, no measure of provided. In fact, the conclusion seems to be that such fitting is inappropriate. I would completely remove figure 3. Alternatively, a new aspect of the software should be introduced.

Response: We agree with the reviewer and the confusion that the term asymmetry has caused in this context.

We have completely revised Figure 3 by showing cell shape orientation (which belongs to the polarity category) and cell elongation (here length to width ratio, which belongs to the morphology category). We applied this to two new conditions including static and high shear stress (20 dyne/cm²).

Inspired by the reviewer's question, we also calculated left-right asymmetry as an additional feature in our pipeline. The changes can also be seen visually in Supplementary Figure S6. While cells at 6 dynes/cm² are very symmetric and well aligned parallel to the flow, there is more variation and more asymmetric shapes at 20 dynes/cm², a feature not captured by either elongation or orientation. However, as the focus here is on orientation and morphology, we have also renamed the results section to 'Shape orientation and morphology'.

As we need to explain to our reader the difference between axial and directed "front-rear" polarity, Figure 3 is crucial. The statistics for axial data are slightly different and are often neglected in the literature. To our knowledge, the polarity index or V-score has not been used in cell biology in the context of axial polarity. Therefore, we believe that this section and Figure 3 are a valuable contribution to this article.

For further clarification: The procedure of fitting an ellipse to the shape is equivalent to calculating moments, but gives a graphical explanation. This is common practice in image analysis and is the standard procedure in well-established Python packages such as scikit-image and other software packages.

8) *In figure 5B, there are some cells in the middle of images that appear not to be included (i.e. they are black whereas all other cells are between shades of blue or red). Why were they excluded? Did they lack a nucleus? Some could be gaps in confluence, but many look very*

cell-shaped. In fact, figure 5A has a gap that definitely has a nuclei, but I will admit that the nucleus looks very large. The authors should better define the exclusion criteria used.

Response: As indicated correctly by the reviewer these are segmentation errors and not actual gaps in the monolayer. Segmentation accuracy is very dependent on image quality and modality, see Stringer et al., 2021 (<https://doi.org/10.1038/s41592-020-01018-x>), meaning there is no model that provides 100% accuracy. All current models will contain segmentation errors. However, we have the ability to train our own model or correct segmentation errors in the pipeline if needed. For reproducibility we did not manually correct for this error, but used the publicly available cellpose “cyto3” deep learning model. If necessary we could provide a manually corrected segmentation mask. We have also added an FAQ to our website pointing to this possibility (<https://polarityjam.readthedocs.io/en/latest/FAQs.html>).

9) Figure 6B defines the 3 junction measures, but it is not clear how these measures are defined. How is the interface defined? How is the fragmented junction area defined? I thought this would be present in S1 but it is not. It is measured in S2, but I still don't understand how these regions are defined.

Response: We agree that these were not easy to understand. We have refined our pictograms in Figure 6B to more clearly illustrate how the feature is calculated. In these pictograms, the sum of intensities is marked with a red sum symbol, and areas are denoted by a red area symbol. Additionally, we updated our naming convention in Supplementary Figure S1C, now referring to the "interface area" and "protein area" for a single cell (previously labelled as "centered membrane mask" and "centered junction mask").

With these quantities we can calculate the interface properties:

- Interface occupancy: “protein area” over “Interface area”
- Intensity per interface area: sum of intensity in the “interface area” over “interface area”
- Cluster density: sum of intensity in the “protein area” over ““protein area”

For the calculation and naming we follow the already published software “JunctionMapper” (see <https://doi.org/10.7554/eLife.45413>).

10) Supplementary table s5 is unnecessary. If you take out the sentence “The categories are shown in table S5.”, then each category is already nicely defined in the main text of appendix 3. It also reads better because you are not distracted by flipped to table S5.

Response: We restructured our representation of features to contain both, the mathematical allocation (e.g. now category - see supplementary table S1), as well as for which biological cell component they can be extracted (former categories, now targets - see supplementary table S2).

Minor

*Figure 1 - flow direction should be indicated in figure not just in legend
There is a typo for "nuclei" .*

Response: We agree with the reviewer and revised Figure 1.

For figures 3A/5A, the stains used are not indicated in the legend. Figure 6 indicates what is blue but not what is yellow.

Response: We added the missing information to the captions and panel in Figure 3A/5A and Figure 6.

What level of shear stress was applied in figure 4, only the duration is indicated.

Response: We thank the reviewer and added the shear stress level, in this case 20 dyne/cm², to the caption.

In the methods, under "Nucleus and organelle displacement", the authors state "The displacement orientation from the nucleus to the centre of mass of the cell can be defined as $\alpha = \sigma \cdot \text{atan2}(y^t - y^r, x^t - x^r)$ ", but what is defined is then actually an alpha for any two measures (organelle-nucleus, nucleus-cell, etc). That first sentence indicates that alpha is only for nucleus-centre of mass. The part of the sentence "from the nucleus to the centre of mass of the cell" is incorrect (or specific to one condition), I believe.

Response: We have generalised polarity features and renamed this part to "Directed 'front-rear' polarity". Indeed, the "target" is now called "front" and "reference" is called "rear", which makes it easier to understand that we define a directed polarity with this approach. Polarity features like nuclei-Golgi polarity, nuclei displacement with respect to cell-centre of mass are just examples. The formula was adapted to the terminology $\alpha = \sigma \cdot \text{atan2}(y^f - y^r, x^f - x^r)$ with the respective indices.

Unless required by the journal, the supplemental tables should come in line with the text. I was reading the methods, which sent me to appendix 3, which then sent me to supp table 5. I did "ctrl-f" and every supplemental table is called only once in the pdf. Therefore, they could easily be placed in line with the text.

Response: We agree with the reviewer and made sure that all Supplementary Figures and Tables are referenced in the main text.

The main text refers to Appendix 3, but there are no appendices, only "Supplementary notes". If the supplementary note is the appendix, please use one term.

Response: We substituted Appendix 3 with Supplementary notes 3.

μ is both the angle for the flow and the central moment. A different symbol should be used for the flow angle.

Response: We replaced the polar direction μ by α_p for a given polar direction (here flow). Note that we - after improvement through the reviewer - use α for directional data and ϕ for axial data. Consistent with polar front-rear direction, which is now described by the symbol α_p , the polar orientation for axial features is given by the symbol ϕ_p .

In the methods, the title of the section is "Nucleus and organelle displacement" but it should be "Nucleus and organelle displacement orientation". An angle not a displacement is defined.

Response: Thank you very much for finding this error. We corrected our text accordingly.

Reviewer #2 (Remarks to the Author):

The manuscript describes a new software tool "polarity-jam" for analyzing directional cell features, e.g. of endothelial cells in blood vessels affected by flow. This software integrates existing cell segmentation tools with a python based pipeline for feature extraction and a web-based tool for feature analysis.

This work could provide a valuable software package for cell feature analysis, as it integrates segmentation with feature analysis for an important application in a user-friendly manner (with a few caveats, see comments on below). The availability of off-the-shelf and generalizable cell segmentation algorithms is increasing the need for such analysis tools that enable standardized downstream analysis for specific applications. As such I believe that this work is suitable for publication in Nature Comms.

Response: We thank the reviewer for giving their detailed assessment.

However, I believe that a few changes are necessary to improve the clarity of presentation and the documentation of the software.

Regarding the manuscript: The manuscript is overall well written and the graphics represent the results well and are easy to understand. The methodology is clearly described and contains enough details for reproducibility. The conclusions drawn are warranted given the presented data.

However, the manuscript does not provide a fully clear overview of the features provided by polarity-jam and which features match based to a user's problem. The first 5 result subsections, "Asymmetris in subcellular localisation of organelles", "Shape asymmetries", "Quantification of intracellular signalling gradients", "Localized marker expression of KLF4" and "Junction morphology" give examples of different analysis that can be performed based on the cell features implemented in polarity-jam, but an easy overview of the different features that are available and their applicability is missing. I suggest to first give a high-level description of the different feature categories in an introductory paragraph of the results and also add a new sub-figure to Fig. 1 that visualizes this overview.

Response:

We acknowledge that Figure 1 did not give an overview of the features of Polarity-JaM, but provided examples. It was therefore not clear which features are available without referring to the supplement or online documentation. To improve clarity, we now provide a high-level overview of feature categories in Figure 1, including localization, polarity, morphology, and intensity, applicable to various targets such as the cell nucleus, organelle, markers, and junction interfaces.

For example, localisation can be applied to the cell, nucleus or organelles. In the same way, we can calculate morphology features for different targets, e.g. the length-to-width-ratio can be extracted from a cell and nucleus.

We added a paragraph to the introduction and revised Figure 1 to include: a) a sketch of an instructive example, b) a high-level feature description, and c) the targets for feature extraction.

A detailed overview is available in Supplementary Tables S1 and S2, and in our revised online documentation (<https://polarityjam.readthedocs.io/en/latest/Features.html>). For each category and single feature we include a brief description, and we provide sketches in the supplementary text of our manuscript and online documentation to facilitate an easy understanding.

Please also indicate the differences between these features clearly, e.g. I am currently unsure what the difference between "nuclei-organelle polarity" and "fluorescent marker polarity and intensity" is, since the organelle / Golgi analysis is also based on a fluorescent marker. Each result subsection should then indicate which of the feature categorie(s) is used.

Response: We thank the reviewer for this comment and suggestion.

In the revised manuscript, we introduce targets to make clear that nuclei-Golgi polarity is calculated from the nuclei (target 1) and the organelle (target 2), while fluorescent marker polarity is calculated from the cell (target 1) and the marker intensity (target 2), where intensity refers to the distribution in the cell.

We changed the text in introduction and results (Section “Asymmetries in subcellular localisation of organelles.”). Furthermore we renamed the former method section “Nucleus and organelle displacement” to “Directed front-rear polarity features” and adapted our notation. In our new notation target 1 would define the “rear” and target 2 the “front”, which allows us to define directed “front-rear” polarity.

Note that in the particular example of *nuclei-organelle polarity* and *fluorescent marker polarity* the difference in terms of targets are as follows:

nuclei-organelle polarity

- Target 1: centroid of nucleus mask
- Target 2: centroid of organelle mask

fluorescent marker polarity and intensity (Feature name: marker nucleus orientation)

- Target 1: centroid of nucleus mask
- Target 2: weighted centroid of marker channel

In principle readouts might provide similar values if the user configures the marker channel to be the organelle channel, however organelle masks (here Golgi) are binary images and their centroids are thus not based on raw intensity values any more.

To clarify which feature is used for each result section in our manuscript, we added pictograms of each feature and target to Figures 2, 3, 4, and 5, and improved the pictogram representation in Figure 6.

Regarding the code and documentation: Overall, I found the code easy to install and to use.

Response: We thank the reviewer to take the time to install and test the usage of our software.

However, the documentation of the code is still fairly shallow. In particular, I did not find any documentation for the napari plugin; the github repo references the wiki where no further information on the plugin can be found. I think that the napari plugin is crucial for visualization purposes to verify the segmentation results in an easy manner, since the reliability of the analysis hinges on a correct segmentation. It would also help computationally less versed users to make use of the plugin.

Response: Indeed, it was not clear how to use the plugin at the point of the review. We adapted our documentation in the installation section <https://polarityjam.readthedocs.io/en/latest/Installation.html#manual-installation-of-the-napari-plugin-for-polarityjam> and thank the reviewer for finding this important shortcoming.

Furthermore, I am also missing a section on "Which feature should I use for my analysis problem" from the wiki. <https://polarityjam.readthedocs.io/en/latest/Features.html> just lists the available features without much further explanation. While I understand that it is not possible to exhaustively address all possible use-cases it would be very helpful to have some examples and guides that explain which features to use given a certain imaging condition. E.g. "I have a cytosol and nuclear stain with additional golgi fluorescent marker. Which features can I use for analyzing reaction of my cells to different directed flow conditions." A few of these examples would clearly help users of the tool.

Response: We agree with the reviewer that a better overview and organisation of the available features is necessary. Hence, we restructured our manuscript and feature documentation to contain both, a feature overview and how to extract features for given targets (e.g. Golgi & nuclear stain).

In this context a walk-through tutorial, either as a video or a page with screenshots, that presents one or more example analyses from start to finish would also help.

Response: We adapted the usage section of our documentation (<https://polarityjam.readthedocs.io/en/latest/Usage.html#napari-plugin>) to include a video of a) the installation process, b) the usage of the napari plugin, as well as c) the usage of the web-app.

Finally, I think it would make sense to add support for custom, already computed, segmentation results for cases where the in-built methods of polarity-jam are not a good fit and users already have a custom segmentation solution.

Response: We fully agree with the reviewer. We offer the option to use custom segmentations from algorithms not yet supported by Polarity-JaM, which is currently described in the API section that targets more advanced users. To make this information more accessible to users less familiar with coding, we have added a Frequently Asked Questions (FAQ) section to our documentation (<https://polarityjam.readthedocs.io/en/latest/FAQs.html>).

In summary: I think this is a valuable and technical sound contribution. Before acceptance I recommend the following improvements (concise summary of the points I made above):

- Provide a concise overview and distinctions of the implemented features at the beginning of Results and in Fig. 1.
- Add documentation for the napari plugin.
- Add documentation that explains the different features with a focus on which features are applicable for a given experimental condition and analysis problem.
- Add tutorials that show a walk-through of using the tool (either screenshots or video, can be combined with the previous point).
- Add support for external segmentations. (This point is not as important as the others, since the tool is still useful without. It may also be possible through the Python API already, which I have not reviewed in much detail. If that is the case maybe just add a short section on this in the doc)

Response: We again thank the reviewer for their valuable feedback that clearly helped to improve the quality of our work. Based on the feedback, we made the following adaptations:

- Updated Figure 1 and the documentation to provide a high-level overview of the supported feature categories and addressable targets.
- Revised the manuscript to align with this new structure.
- Added both video and text-based documentation for the Napari plugin, which also serves as a walkthrough of the tool's workflow, including web-app usage.
- Created a Frequently Asked Questions (FAQ) section in the documentation to explain how to use custom segmentations and address other common queries.

Details: Figure 1 is missing "A", "B" as well as the black arrow that should indicate the flow direction according to the caption.

Response: Thank you for pointing us to this important detail. We adapted our figure accordingly.

Reviewer #2 (Remarks on code availability):

See main comments.

Polarity-JaM Manuscript: Point-by-Point Response to Reviewers

Reviewer #1 (Remarks to the Author):

In the revised version, the authors have made significant improvements that help with the accuracy and readability of the article as well as adding several features that will help other labs get up and running with the application.

I still have two very minor points, but important. Because, within the introduction paragraphs there is one statement that is technically incorrect and a second statement that is absolutely incorrect.

1) The authors refer to “fluid shear stress”. It is actually a wall shear stress. Shear stress is $\mu \cdot dv/dr$. dv/dr has a value at all values of the radius, even in the fluid. But endothelial cells sense the shear stress at $\mu \cdot dv/dr$ at $r=R$. And this is the wall shear stress. See figure 2 of https://www.homepages.ucl.ac.uk/~uceseug/Fluids2/Notes_Viscosity.pdf. It defines both the shear stress at a random y (which is a fluid shear stress) and at $y=0$ (which is the wall shear stress). I realise this is commonly called “fluid shear stress” by biologist, but that’s actually incorrect. The whole discussion can be avoided by simply saying “ECs are sensitive to shear stress when blood flow passes it runs through blood vessels”. I would therefore advise the authors to remove the word “fluid”.

Response: Thank you for this comment, we fully agree and have removed the word “fluid”.

2) “Collective cell orientation was also correlated with both types of shear flow, pulsatile and laminar, but pulsatile flow does not organise orientation direction as strongly as laminar flow”.

This sentence does not make any sense. Pulsatile flow is laminar. I think the authors mean steady flow versus pulsatile. I realise this wording existed in the author’s first submission, but I missed it. The references for this statement also do not make sense. Ref 17 is about valvular endothelial cells (Nandini et al). Valvular endothelial cells are special, they align perpendicular to flow rather than parallel and show other unique behaviours. But Ref 17 is also a very bad paper. They use a soft matrix when a valve is cartilage and actually quite stiff. They never identify the source of valvular endothelial cells (especially concerning since their “endothelial cells” express α SMA). They never define their pulsatile wave form or the frequency of the paper pulsation. Plus, they show the opposite of the author’s sentence. They show greater loss of circularity, greater increase in aspect ratio at 48h pulsatile than 48h “laminar”. The second reference for this statement (Ref 18, Vion et al.) does not look at pulsatile flow. Therefore, two things are needed here. First, it must say steady versus pulsatile, not laminar versus pulsatile. And second, other references are needed. But the

authors will find that the vast majority of papers find that pulsatility does the same thing as steady flow, just faster and more pronounced. Though perhaps the authors meant “disturbed” versus “laminar”?

Response: We are grateful for this detailed and insightful feedback. We understand the confusion and have clarified that we were comparing "undisturbed" versus "disturbed" flow, not "laminar" versus "pulsatile" flow. The revised sentence now accurately reflects this comparison. Therefore, we replaced this sentence with "ECs exposed to undisturbed flow are elongated with increased LWR and aligned with the direction of flow, whereas ECs in areas of disturbed flow are more cuboidal and randomly oriented, both in vivo and in vitro (Ref. 17, Dessalles et al. 2021)" and removed the references Nandini et al. and Vion et al. in this context and sentence.

*And there are two typos:
Alignement on p7 should be alignment.
proteing should be protein*

Response: We thank the reviewer for the careful reading, this has been corrected.

Reviewer #2 (Remarks to the Author):

*The revision has addressed (almost) all the points I have raised in my initial review:
- Figure 1 and the introduction now provide a much better overview of the available features and explains their differences.
- The documentation of the tool has improved a lot, including documentation of the napari plugin with a video.
- Several other minor points were addressed.
- The only point not fully addressed are more concrete use-cases / examples in the documentation, which could be quite helpful for potential users. This can however be easily added post publication, also based on experience with users of the software.
I thus recommend to accept the manuscript as is.*

Reviewer #2 (Remarks on code availability):

The code is now well documented.

Response: We thank the reviewer for the thorough review and positive comments on our revisions. We are glad our efforts to improve the introduction, Figure 1, and documentation have been successful. We will add more use-cases to the documentation post-publication, as suggested. We are pleased that the reviewer finds the code to be well-documented and recommends accepting the manuscript as is.